# KGNER: Improving Chinese Named Entity Recognition by BERT Infused with the Knowledge Graph

**Weiwei Hu** [1] , **Liang He** [1,2,*], **Hanhan Ma** [1], **Kai Wang** [3] **and Jingfeng Xiao** [3]

1   College of Information Science and Engineering, Xinjiang University, Urumqi 830046, China; h_inspire@stu.xju.edu.cn (W.H.); mhh0113@foxmail.com (H.M.)
2   Department of Electronic Engineering, Tsinghua University, Beijing 100084, China
3   State Grid Xinjiang Electric Power Co., Ltd., Urumqi 830000, China; wangkai@xj.sgcc.com.cn (K.W.); xiaojingfeng@xj.sgcc.com.cn (J.X.)
*   Correspondence: heliang@mail.tsinghua.edu.cn

**Abstract:** Recently, the lexicon method has been proven to be effective for named entity recognition (NER). However, most existing lexicon-based methods cannot fully utilize common-sense knowledge in the knowledge graph. For example, the word embeddings pretrained by Word2vector or Glove lack better contextual semantic information usage. Hence, how to make the best of knowledge for the NER task has become a challenging and hot research topic. We propose a knowledge graph-inspired named-entity recognition (KGNER) featuring a masking and encoding method to incorporate common sense into bidirectional encoder representations from transformers (BERT). The proposed method not only preserves the original sentence semantic information but also takes advantage of the knowledge information in a more reasonable way. Subsequently, we model the temporal dependencies by taking the conditional random field (CRF) as the backend, and improve the overall performance. Experiments on four dominant datasets demonstrate that the KGNER outperforms other lexicon-based models in terms of performance.

**Keywords:** named-entity recognition; knowledge graph; conditional random field

## 1. Introduction

Named-entity recognition (NER) is devoted to locating and classifying certain occurrences of words or expressions in unstructured text into predefined semantic categories, such as person names, locations, organizations, etc. This is not only an important upstream task of natural language processing (NLP), but also an essential prerequisite for other related tasks, such as information retrieval [1], relation extraction [2,3], question-and-answer (Q&A) systems [4] and other applications. It has also drawn the attention of the academic community in recent decades.

Chinese language has the characteristic that text consists of characters rather than words, resulting in Chinese sentences lacking clear word boundaries. This also adds new opportunities and challenges to the task of Chinese named-entity recognition. Previous methods have shown that character-based approaches perform better than word-based approaches in Chinese NER, because they are not affected by Chinese word segmentation errors [5,6]. The application of lexical features enables external lexical information to enhance the training of NER [7–10]. However, due to the flexibility of named entity recognition, there may be a large number of out of vocabulary (OOV) named entities in the open domain, which poses a great challenge. In addition, named entities may be ambiguous. For example, for the sentence *"Blue Moon tops Premier League with 12-game winning streak."* The term "blue moon" literally means that the moon is blue in color. However, in soccer news, it often stands for the English Premier League team Manchester City F.C. Nevertheless, knowledge graphs containing domain knowledge may be helpful in this regard [11,12].

The rise in pre-trained models in recent years has brought new solutions to named-entity recognition. Many NLP tasks have shown promise for unsupervised pre-trained language representation models, e.g., BERT [13] and ELMo [14]. After being pre-trained on many unlabeled data to obtain generic representations, the pre-trained models are often equipped with back-end models to suit downstream tasks. Although successful results have been achieved for limited labeled data in a specific domain, these models often perform poorly on knowledge-driven tasks.

Knowledge graphs (KG) represent entities and relationships in a graph and contain a wealth of information regarding world knowledge. Therefore, they are an important complement to existing pre-trained language models and have the potential to address the sparsity problem existing among most NLP tasks. Recall the previous example. The term "blue moon" also implies the Manchester City team in the soccer domain. We can infer that, in order to make the model aware of this hidden meaning, we should empower it with the ability to inquire relevant information from reliable knowledge sources. The knowledge graph opens the door to this information, allowing details about entities or relations, which may never have been encountered in the training data, to be learned.

There are two lines of recent methods enhancing knowledge-based neural Chinese named-entity rcognition. The first directly considers integrating knowledge when we begin to train a model such as ERNIE [15]. Although domain-specific knowledge graphs can be injected in the pre-training phase, this training process can be expensive and time-consuming. The second considers the integration of domain-specific knowledge into a pre-trained model such as K-BERT [12]. However, there are two challenges to face in the road of knowledge integration:

1. Knowledge noise (KN), disturbing knowledge, is often incorporated into the modeling, which may confuse semantic information.
2. Heterogeneous information fusion (HIF), word embeddings in text, and entity embeddings in a knowledge graph are obtained in different ways, and they are two independent vector spaces.

To cope with the above problems and challenges, this paper proposes KGNER based on BERT and KG to extract the information of entities and enable language models to obtain detailed information, beyond the training data. There are three main contributions of this paper, which are summarized as follows:

- This paper proposes a new position coding method that can make good use of the detailed information of the knowledge graph and also preserve the original sentence semantic information.
- Our method avoids KN and HIF problems during the process of injecting structural information in knowledge graph.
- We adopt a conditional random field model for better modeling of sequential information.

To verify our proposed method, we conducted elaborate experiments on four publicly available datasets, and the results demonstrate the effectiveness of KGNER.

The rest of the paper is organized as follows: We first present related work in Section 2. We then describe the proposed methods and formulations in details in Section 3, followed by experiments and results in Sections 4 and 5. We conclude our paper with a discussion on future work in Section 6.

## 2. Related Work

### 2.1. Named-Entity Recognition

The named-entity recognition task was conventionally formulated as a sequence labeling problem, where entity boundaries and category labels are jointly predicted. Compared with English, Chinese does not have apparent word boundary features, but it is critical to utilize word boundaries and semantic information in Chinese NER. Usually, before proceeding with NER, word segmentation is performed. However, word segmentation can suffer from error propagation [16,17]. It has been shown that character-based methods are superior to word-based methods for Chinese NER. However, one disadvantage of

character-based methods is that they do not take full advantage of explicit word and word sequence information. Word information plays an essential role in Chinese NER. To overcome this limitation, many efforts have been devoted to incorporating word information by leveraging lexicon features. Zhang et al. [8] matches sentences with dictionaries to form a lattice structure that integrates potential word information into character-based LSTMs. It shows that fusing lexical information into the native LSTM may be helpful for NER.

However, there are some problems with the above-mentioned method. The RNN structure used by Lattice LSTM cannot utilize the global information, which leads to possible lexical information conflicts. In addition, it is tough to migrate the Lattice-LSTM structure to other neural network architectures. This not only limits its inference speed but also makes it difficult to execute training and inference in parallel. Gui and Ma et al. [9] addressed this problem by modeling characters with potential words in parallel, as well as employing a rethinking mechanism. Hence, their method is more efficient and effective. Ma et al. [18] proposed encoding the matched words obtained from the lexicon into the representations of characters. His model is also compatible with any suitable neural architectures without redesign. Mengge et al. [19] proposed a fresh lattice Transformer encoder with the help of a porous mechanism for Chinese NER, which is able to manipulate lattices in batch mode and catch dependencies between characters and matched lexical words. Gui and Sui et al. [10,20] resorted to a lexicon and character sequence to build a graph, transforming NER into a node classification work. However, due to NER's forceful alignment between label and input, their model is compelled to embrace an RNN module for encoding. Li et al. [21] utilized a Transformer that adopts a fully connected self-attentive mechanism to capture long-range dependencies, and also uses position encoding to fuse lattice structures. These methods, based on character-attached lexical information augmentation, have proven to be of great benefit in exploiting lexical information and avoiding word segmentation error propagation, achieving decent results.

### 2.2. Pre-Trained Language Models

In terms of deep learning, in order to possess a deeper model, we need to feed it adequate labeled data. In the NLP domain, labeled data are an expensive resource. Pre-Trained Models (PTMs) that were pre-trained from a great deal of unlabeled data have led to prominent performance gains in many NLP tasks. Two major paradigms are summarized: shallow word embedding and pre-trained encoders.

**Shallow word embedding**: This class of PTM paradigms is what we usually call "word vectors", whose main feature is learning context-independent static word embedding, which is mainly represented by Word2vec (CBOW [22], Skip-Gram [22]), Glove [23], etc. The main drawback of shallow word embedding is that word embedding is context-independent, and the embedding vector of each word is always the same. Therefore, it cannot solve the problem of multiple word meanings.

**Pre-trained encoders**: The second class of PTM paradigms is the pre-trained encoder, which overcomes the issue of multiple word meanings by yielding a vector of contextually relevant words. This class of pre-trained encoders output vectors is called "context-sensitive word embedding". The representatives of this class are ELMo, GPT, Bert, etc. To solve the problem of multiple-meaning words, Peters et al. [14] proposed using ELMo to distinguish the semantics of words using the semantics of the context. A two-stage process of pre-training and feature extraction is used. However, ELMo employs a tandem bidirectional RNN to obtain the semantics of the context. GPT [24], proposed by the OpenAI team in 2018, has a similar idea to ELMo in that both use a two-stage feature extraction process. The difference is that GPT replaces the RNN in the second stage using a Transformer with a better feature extraction ability to better capture long-range linguistic structures, but uses a one-way language model that can only accept semantic information from the above context.

Google proposed the BERT model in late 2018 [13], which uses precisely the same two-stage structure as GPT: firstly, language model pre-training; secondly, solving downstream tasks using a Fine-Tuning model. The BERT model mainly addresses the problem that

the GPT model does not use a bidirectional language model in the pre-training stage. Unlike previous language representation models, BERT is aimed at pre-training deep bidirectional representations by jointly adjusting the left and proper contexts in all layers. The Transformer Encoder that was used internally can capture the semantic information of a textual context using the self-attention mechanism. The word vectors obtained using the BERT model have better semantic properties than language models such as Word2vec. In generic domains, the *F*1 value has reached a high score of 93.09 on the named-entity recognition task of CoNLL2003. Yang et al. [25] added a softmax to BERT to achieve a state-of-the-art performance on Chinese word segmentation(CWS). Some scholars [26,27] showed that the model using character features of BERT significantly outperformed the static embedding-based approach on Chinese NER and Chinese part of speech (POS) tags. The methods in Section 2.1 integrate discrete and structured knowledge by designing different neural network structures. Liu et al. [28] proposed Lexicon-Enhanced BERT (LEBERT) for Chinese sequence labeling, which causes BERT layers to consume the external lexicon by means of a Lexicon Adapter layer. Compared with existing approaches, their model promotes the fuller integration of lexical knowledge at the shallow level of BERT.

In the last two years, generative pre-training models have also been heavily studied in addition to the above two pre-training paradigms. Lewis et al. [29] proposed Bidirectional and Auto-Regressive Transformers (BART). Colin et al. [30] proposed Unified Text-to-Text Transformer's T5. They absorbed the features of BERT's bidirectional encoder and GPT's left-to-right decoder and built on the standard sequence-to-sequence Transformer model. They are more suitable for text generation scenarios than BERT. They also have more bidirectional contextual information compared to GPT. Du et al. [31] proposed "All NLP Tasks Are Generation Tasks". Yan et al. [32] used BART to solve three types of NER tasks, and Cui et al. [33] used BART to solve how to transfer richer domain knowledge to a sparse knowledge domain, and all of them achieved good results.

*2.3. Knowledge Graph*

A knowledge graph can be understood as a semantic network whose contents reveal specific relationships between entities. It is often used to provide a structured and visual description of things that exist in the real world and their interrelationships. Today's knowledge graphs can be used to refer to various large-scale visual knowledge bases in general. The triad form is generally used as a generalized representation of knowledge graphs, i.e., G(E, R, S), where $E = \{e_1, e_2, ..., e_n\}$ denotes the set of entities in the knowledge base, which contains $|E|$ different entities; $R = \{r_1, r_2, ..., r_n\}$ is the set of relations in the knowledge base, containing $|R|$ different relations; $S \subseteq E \times R \times E$ represents the set of triples in the knowledge base. Usually, the relationship between entities is defined as a semantic predicate, so the representation also occurs in the form of a Subject, Predicate, Object (SPO) triad. The basic form of the triad mainly includes (Entity1, Relationship, Entity2) and (Concept, Attribute, Attribute Value), etc. The entity is the most basic knowledge element in the whole knowledge graph, and different relationships exist between different entities. Knowledge graphs contribute to various applications, from search engines to question-answering systems.

In recent years, a number of large-scale, open-domain knowledge graphs have been made available to the public, such as CN-DBpedia [34], HowNet [35], and also domain-specific knowledge graphs, such as MedicalKG [12], have been developed for medical care. Researchers are also actively exploring how to integrate knowledge graphs into NLP tasks. TransE [36] and other KG embedding approaches have been described for encoding the entities and relations into numerical representations. Some scholars attempt to learn entity/relation embeddings together with their semantic information [37,38], while some strategies focus on graph structure encoding [36,39–41]. Then, certain NLP jobs will profit from the use of these graph embeddings. In order to classify the text, Annervaz et al. [42] suggested a knowledge-graph-augmented neural network in which the context vector (via LSTM, which encapsulates the complete input text) is combined with the entity and

relation vectors that are obtained from the database. Knowledge attention is stated as a function of the entity that is to be typed. The limitation lies in the application of a recurrent architecture, resulting in time inefficiency and highly computational waste. Xin et al. [43] proposed a novel attention mechanism that leverages information from knowledge bases (KB) and jointly takes text and KB into consideration. He et al. [11] proposed a new word representation for named-entity recognition by encoding entity information (from an external knowledge base) through a new gated recurrent unit (GRU) and by modeling the relation context between entities through a new attention function.

To use knowledge graph embeddings (KGE), we observed that the previously proposed approaches mainly used pre-trained static embeddings acquired from considerable sources such as Wikidata. KGE in these models is radically dependent on the assumption that the tail entity is a linear transformation of the head entity as well as the relationship, making them non-contextualized in nature. Therefore, the incorporation of knowledge in pre-trained models to obtain performance gains in NLP tasks has received considerable attention in recent years. ERNIE [15,44] combined pre-trained entity embedding in knowledge graphs with corresponding entity mentions in a text to enhance the textual representation. To solve the HIF problem, entity vectors and text representations are fused by nonlinear transformations at locations where entity inputs are available to fuse lexical, syntactic, and knowledge information. K-Adapter [45] is capable of injecting multiple knowledge by independently training different adapters for diverse pre-training tasks. K-BERT [12] explicitly injects relevant triples extracted from KG into sentences to obtain extended tree inputs for BERT. However, as the added knowledge increases, the sentence length inevitably increases. If the sentence length exceeds the model's maximum capacity, the original semantic information of the sentence will changed This defeats our original purpose. Fortunately, our approach overcomes this problem very well.

## 3. Model

### 3.1. Model Architecture

Formally, given a knowledge graph and a Chinese sentence with $n$ characters $s_c = \{c_1, c_2, ..., c_n\}$, where $c_i$ denotes the $i$-th character in the sentence and $n$ is the length of this sentence. KG, denoted as $\mathbb{K}$, is a collection of triples $\varepsilon = (w_i, r_j, w_k)$, where $w_i$ and $w_k$ are names of entities. Each token $w_i$ is included in the vocabulary, $w_i \in \mathbb{V}$ and $r_j$ is the relation between $w_i$ and $w_k$. All the triples are included in KG.

As illustrated in Figure 1, the model architecture is made up of three modules, i.e., the knowledge layer, embedding layer and output layer.

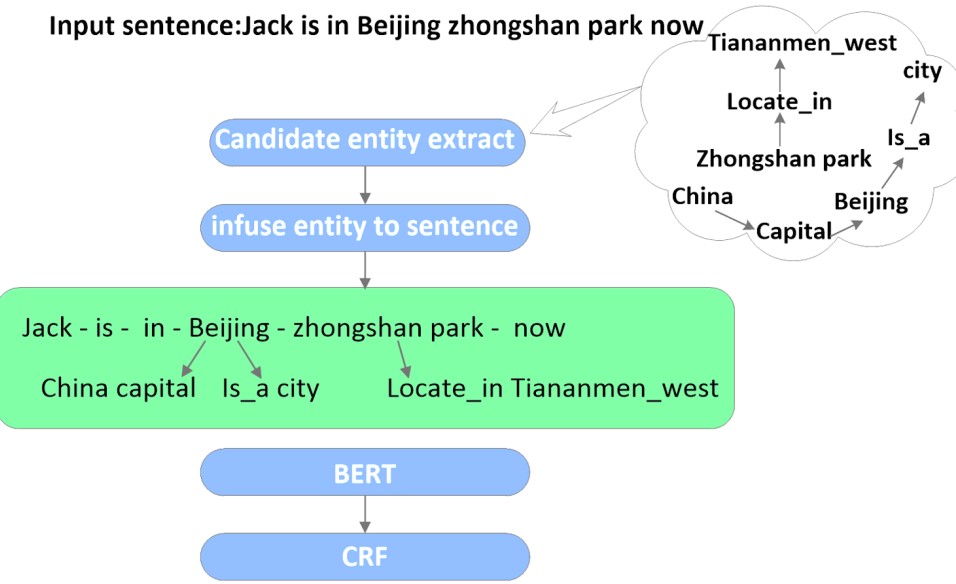

**Figure 1.** The model structure.

### 3.2. Knowledge Layer

The knowledge layer (KL) is designed to inject the knowledge of the knowledge graph into the sentence. Specifically, given an input sentence $S = \{w_0, w_1, w_2, ..., w_n\}$ and a KG, we can obtain a sentence tree. This process can be broken down into two steps: knowledge query (K-Query) and knowledge injection (K-Inject).

In K-Query, first, the sentence needs to be cut apart, and the knowledge graph is built as a look table. Each token in the sentence is taken to match with the look table for selecting the triples. K-Query can be formulated as (1).

Next, to create a sentence tree $T$ with a wealth of knowledge. K-Inject places the triples in $E$ to their corresponding position so that the triples queried can be injected into the original sentence $S$. The structure of $T$ is illustrated in Figure 2, where $E = \{(w_i, r_{i0}, w_{i0}), ..., (w_i, r_{ik}, w_{ik})\}$ is a collection of the corresponding triples. K-Inject can be formulated as (2)

$$E = K\text{-}Inject(S, E) \tag{1}$$

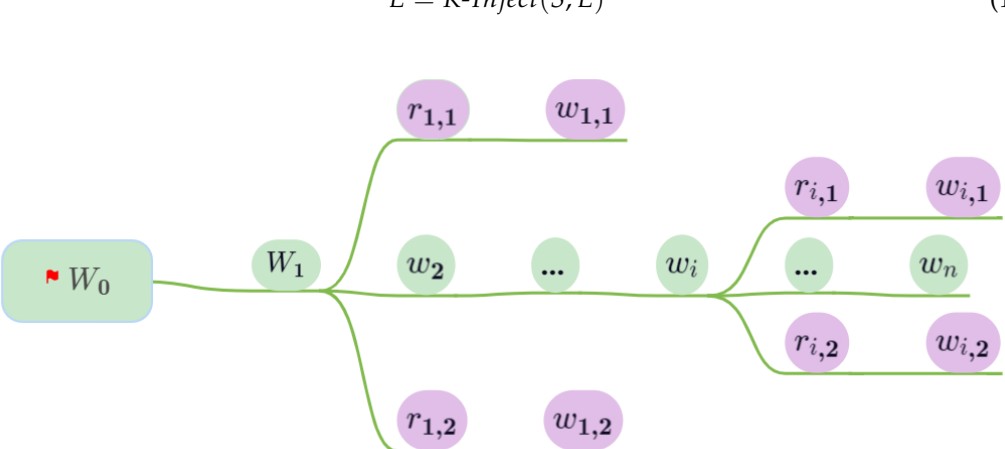

**Figure 2.** Structure of the sentence tree.

### 3.3. Embedding Layer

In the embedding layer, the BERT encoder can encode sentence trees as embedding representation. The embedding representation of BERT consists of token embedding, position embedding, and segment embedding. There is one significant difference between our model input and the general BERT. Our model input is a sentence tree instead of a token sequence. Therefore, it is crucial for BERT to convert the sentence tree into a sequence while preserving the original information of the sentences.

**Token embedding**: For this work, our token embedding is in line with BERT. We adopt the vocabulary provided by Google BERT in this paper. Each token in the sentence tree is turned into an embedding vector of dimension $H$ with the help of a trainable lookup table. Moreover, $[CLS]$ is considered as as a category tag and $[MASK]$ is regarded as a masked token in BERT. Nevertheless, tokens in the sentence tree are expected to be rearranged before the embedding operation is performed.

**Seg embedding**: When multiple sentences are entered, BERT uses segmentation embedding to distinguish between the different sentences. In this work, we only use a sentence. $\{A, A, A, A, ..., A\}$, a sequence of segment tags, is adopted to mark a sentence.

**Position embedding**: For BERT, the position embedding contains all the structural information of the BERT's input sentence. Without position embedding, this will be treated as a bag-of-word model, resulting in a lack of structural information (i.e., the order of tokens). We are allowed to add the missing structured information to the unreadable rearranged sentence.

$$E = K\text{-}Query(S, \mathbb{K}) \tag{2}$$

**Mask-Self-Attention**: The risk involved in using knowledge is that the original sentence can suffer from changes in meaning. To avoid affecting the meaning of the original sentences, some measures need to be taken. First, a sentence tree is constructed by knowledge, and each word in the sentence tree is encoded by absolute position. Then, the tokens in the sentence tree are flattened into a sequence of the token by means of their absolute-position index. In other words, the original sentence is followed by tokens in the branch. As shown in Figure 3, the sentence tree is rearranged as "Jack is in Beijing Zhongshan park now capital China is_a city locate_in Tian'an'men west". The advantage of this is that the original semantic information of the sentence is preserved, but the sentence is still unreadable. For example, there is no connection between [*now*] and [*capital*].

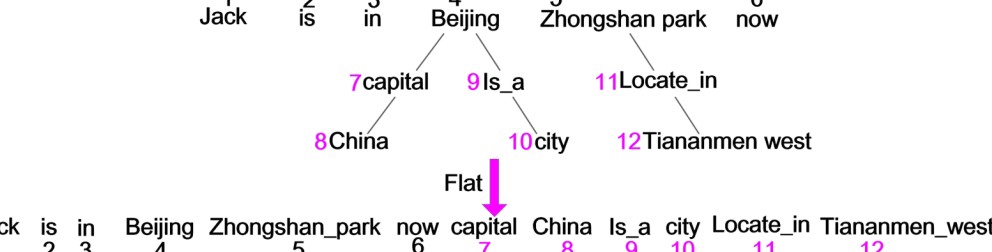

**Figure 3.** Flat of the sentence tree.

Fortunately, this can be solved by limiting the visible area of each token by the mask matrix. We introduce the visible mask matrix in Figure 4. The sentences of different branches are invisible to each other. Here is an example to explain this. [*China*] and [*city*] are not visible to each other; however, [*Beijing*], [*capital*] and [*China*] are visible each other. In this way, the words in the sentence not only acquire the corresponding knowledge information but also prevent the influence of irrelevant words. The visible matrix is defined as (3).

$$M_{ij} = \begin{cases} 0 & w_i \ominus w_j \\ -\infty & w_i \oslash w_j \end{cases} \tag{3}$$

where $w_i \ominus w_j$ indicates that $w_i$ and $w_j$ are in the same branch, while $w_i \oslash w_j$ are not. $i$ and $j$ are the absolute position index. To avoid the semantic changes caused by taking advantage of the sentence structure information in $M$, we utilize a mask-self-attention, which is an extension of self-attention. Formally, the mask-self-attention is defined as (4).

$$Q^{i+1}, K^{i+1}, V^{i+1} = h^i W_q, h^i W_k, h^i W_v, \tag{4}$$

$$S^{i+1} = \text{softmax}\left( \frac{Q^{i+1} K^{i+1^\top} + M}{\sqrt{d_k}} \right), \tag{5}$$

$$h^{i+1} = S^{i+1} V^{i+1} \tag{6}$$

where $W_q$, $W_k$ and $W_v$ are trainable model parameters. $h^i$ is the hidden state of the $i$-th mask-self-attention blocks. $d_k$ is the scaling factor.

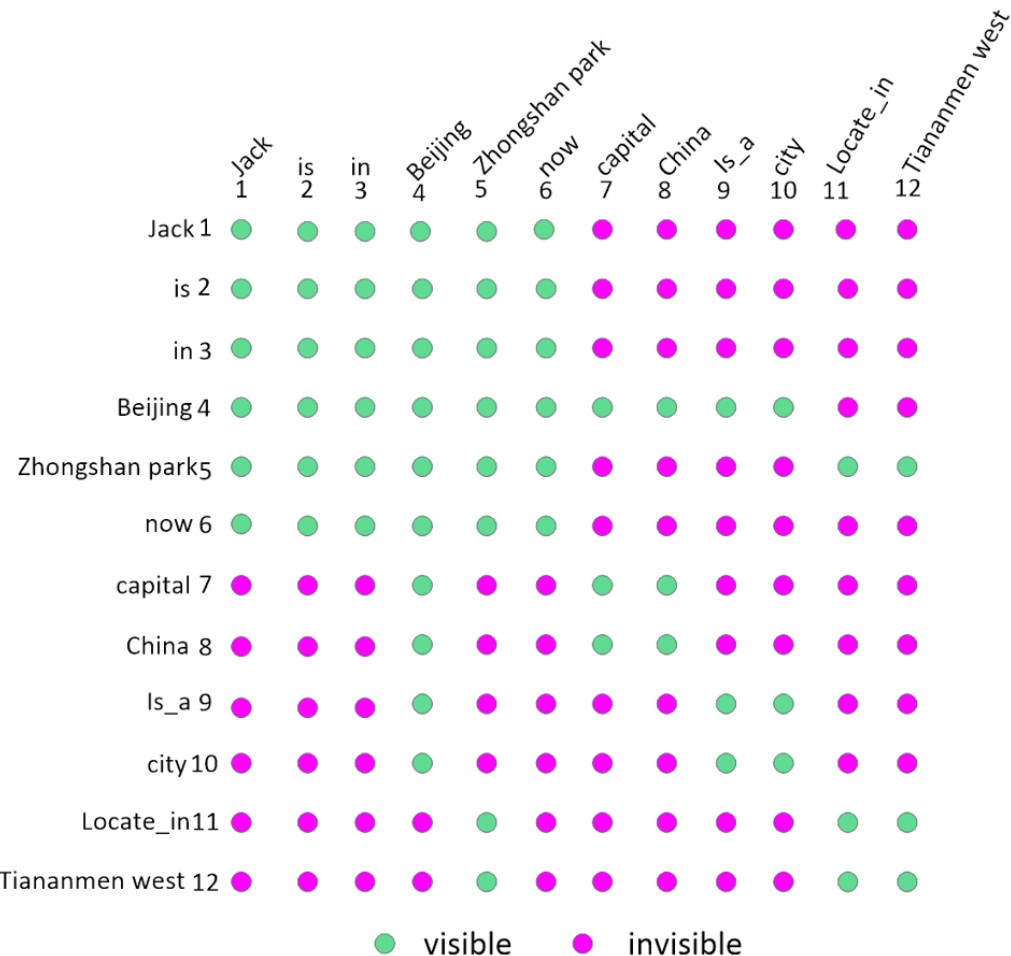

**Figure 4.** Mask matrix.

*3.4. Output Layer*

For sequence labeling tasks, there is a strong connection between the labels. For example, compared to a verb, an noun is more likely to follow a adjective. In NER, using a standard BIO annotation [46], I-PER cannot be followed by I-ORG. Therefore, considering the connection between adjacent tags can be of great benefit to NER. As conditional random field (CRF) [47,48] can make full use of the neighbor tag information when predicting current tag, it can jointly decode the best chain of labels for a given input sentence. Consequently, we consider CRF when modelling a label sequence instead of decoding each label independently.

Formally, we denote a generic input sequence by $z = \{z_1, ..., z_n\}$. $z_i$ represents the input vector of the $i$-th word. We denote a generic sequence of labels for z by $\boldsymbol{y} = \{y_1, \cdots, y_n\}$. $\mathcal{Y}(z)$ represents the set of probable label sequences for $z$. Of all possible label sequences $y$ given $z$, the probabilistic model for sequence CRF defines a family of conditional probability $p(y \mid \mathbf{z}; \mathbf{W}, \mathbf{b})$. It takes the following form:

$$p(\boldsymbol{y} \mid \mathbf{z}; \mathbf{W}, \mathbf{b}) = \frac{\prod_{i=1}^{n} \psi_i(y_{i-1}, y_i, \mathbf{z})}{\sum_{y' \in \mathcal{Y}(\mathbf{z})} \prod_{i=1}^{n} \psi_i\left(y'_{i-1}, y'_i, \mathbf{z}\right)} \tag{7}$$

$\psi_i(y', y, \mathbf{z}) = \exp\left(\mathbf{W}_{y',y^T}^T \mathbf{z}_i + \mathbf{b}_{y',y}\right)$ are potential functions. $\mathbf{W}_{y',y^T}^T$ denotes the weight vector, and $\mathbf{b}_{y',y}$ denotes bias corresponding to label pair $(y', y)$.

We employ the maximum conditional likelihood estimation to train CRF. Given a training set $\{(\mathbf{z}_i, \boldsymbol{y}_i)\}$, the logarithm of the likelihood (a.k.a. the log-likelihood) is defined by the following equation:

$$L(\mathbf{W}, \mathbf{b}) = \sum_i \log p(y \mid \mathbf{z}; \mathbf{W}, \mathbf{b}) \tag{8}$$

The parameters maximizing the log-likelihood $L(\mathbf{W}, \mathbf{b})$ will be choosed by maximum likelihood training. Decoding aims to find the label sequence $y^*$ possessing the highest conditional probability:

$$y^* = \underset{y \in \mathcal{Y}(\mathbf{z})}{\mathrm{argmax}} p(\mathbf{y} \mid \mathbf{z}; \mathbf{W}, \mathbf{b}) \tag{9}$$

For a sequence CRF model, the Viterbi algorithm is adopted for training and decoding.

## 4. Experiments Setup

An extensive set of experiments were carried out to investigate the effectiveness of knowledge graphs across different domains. In addition, we aimed empirically compare lexicon-based and knowledge-based Chinese NER in different settings.

### 4.1. Data

Four Chinese NER datasets were used to evaluate our model, including (1) **Ontonotes 4.0** [49] (2) **MSRA** [50] (3) **Resume** [8] (4) **Weibo** [51,52]. OntoNotes and MSRA datasets belong to the field of news. Weibo NER dataset was obtained from the social media website Sina Weibo. Resume NER dataset comprised the resumes of senior executives, which were annotated by [8]. Dataset statistics are demonstrated in Table 1.

**Table 1.** Statistics of datasets.

| Dataset | Type | Train | Dev | Test |
|---------|------|-------|-----|------|
| Ontonotes | Sentence | 15.7 k | 4.3 k | 4.3 k |
| | character | 491.9 k | 200.5 k | 208.1 k |
| MSRA | Sentence | 46.4 k | - | 4.4 k |
| | character | 2169.9 k | - | 172.6 k |
| Weibo | Sentence | 1.4 k | 0.27 k | 0.27 k |
| | character | 73.8 k | 14.5 k | 14.8 k |
| Resume | Sentence | 3.8 k | 0.46 k | 0.48 k |
| | character | 124.1 k | 13.9 k | 15.1 k |

### 4.2. Metrics

Precision *P*, recall *R* and *F*-measure were used as measures, defined as follows:

$$P = \frac{N_m}{N_p}, R = \frac{N_m}{N_r}, F1 = \frac{2 \times P \times R}{P + R} \tag{10}$$

$N_m$, $N_p$ and $N_r$ denote the total number of matched entities, predicted entities and real entities, respectively. *F1* is the reconciled average metric of accuracy and recall, and is a composite metric that balances the impact of accuracy and recall.

### 4.3. Hyperparameters

Our BERT parameters are consistent with Google BERT [13]. *L* is the number of mask-self-attention layers, and *A* is the size of each head. *H* is denoted as the hidden dimension of embedding vectors. The model is configured as follows. *L*,*A* and *H* are 12, 12 and 768, respectively. The total trainable K-BERT parameters are the same as BERT(110M). The Adam optimizer with an initial learning rate of $2 \times 10^{-5}$ is adopted. The number of maximum epoch number is 5 for training on all datasets. We set the max length of the sequence to 256 and the training batch size to 16 for all datasets.

*4.4. Baselines*

To verify the validity of the proposed model, we conducted a comparision with the K-BERT [12] in the experiments.

The experimental results on Chinese NER datasets are given in Table 2. In the first part of the table, the first four rows [8,53–55] displayed the performance of lexicon-enhanced, character-based Chinese NER models. They built dictionaries from pre-trained word vectors. The last two rows [18,21] in the same block were the state-of-the-art models, and they integrated lexicon information and BERT using a shallow fusion layer; the medial five rows employed the pre-trained language model. BERT directly fine-tuned a pre-trained ChineseBERT on Chinese sequence labeling tasks. ERNIE [15] extended the BERT by using an entity-level mask to guide pre-training. ZEN [56] explicitly injected N-gram information into BERT through extra multi-layers of N-gram Transformer encoder and pre-training. To integrate lexicon features into BERT, LEBERT came up with a fresh method for Chinese sequence labeling, which directly used a Lexicon Adapter to integrate lexicon information between Transformer layers in BERT. It has a deep integration of dictionaries and BERT. The last two rows improved Chinese NER by BERT infused with the knowledge graph. As we can see, the models based on the knowledge achieved a better performance than lexicon-enhanced models.

**Table 2.** Four datasets results (*F*1).

| Model | Weibo | Ontonotes | MSRA | Resume |
|---|---|---|---|---|
| Zhang and Yang (2018) [8] | 63.34 | 75.49 | 92.84 | 94.51 |
| Zhu and Wang (2019) [53] | 59.31 | 73.64 | 92.97 | 94.94 |
| Liu et al. (2019) [54] | 65.30 | 75.79 | 93.50 | 94.49 |
| Ding et al. (2019) [55] | 59.50 | 75.20 | 94.40 | - |
| Ma et al. (2020) [18] | 69.11 | 81.34 | 95.35 | 95.54 |
| Li et al. (2020) [21] | 68.07 | 80.56 | 95.46 | 95.78 |
| BERT [13] | 67.27 | 79.93 | 94.71 | 95.33 |
| BERT+Word | 68.32 | 81.03 | 95.32 | 95.46 |
| ERINE [15] | 67.96 | 77.65 | 95.08 | 94.82 |
| ZEN [56] | 66.71 | 79.03 | 95.20 | 95.40 |
| LEBERT [28] | 70.75 | 82.08 | 95.70 | 96.08 |
| KBERT [12] | 70.00 | 82.00 | 95.50 | 96.20 |
| KGNER | **71.90** | **82.10** | **95.90** | **96.40** |

## 5. Overall Results

*5.1. Ablation Studies on the Four Datasets*

To study the role of each part of KGNER, we performed ablation experiments on the four datasets and displayed the results in Figure 5. The results demonstrate that the model's performance is declined if the mask matrix is dropped. For example, Weibo is severely injured by 4.4 without a mask matrix. All the tokens are visible in this situation, and some tokens can be interfered by other tokens, indicating that the mask matrix plays an irreplaceable role in the graph structure.

To better show the advantage of our model, we dropped the position coding and mask matrix and simplified the structure. KGNER is equivalent to the BERT without the position coding and mask matrix. The results show that the KGNER reaches a *F*1 score by 0.9, on average, than the BERT on the four datasets. From this discovery, we deduce that the knowledge graph plays an essential role in KGNER. It also manifests that the KGNER has a more robust ability to model sentences.

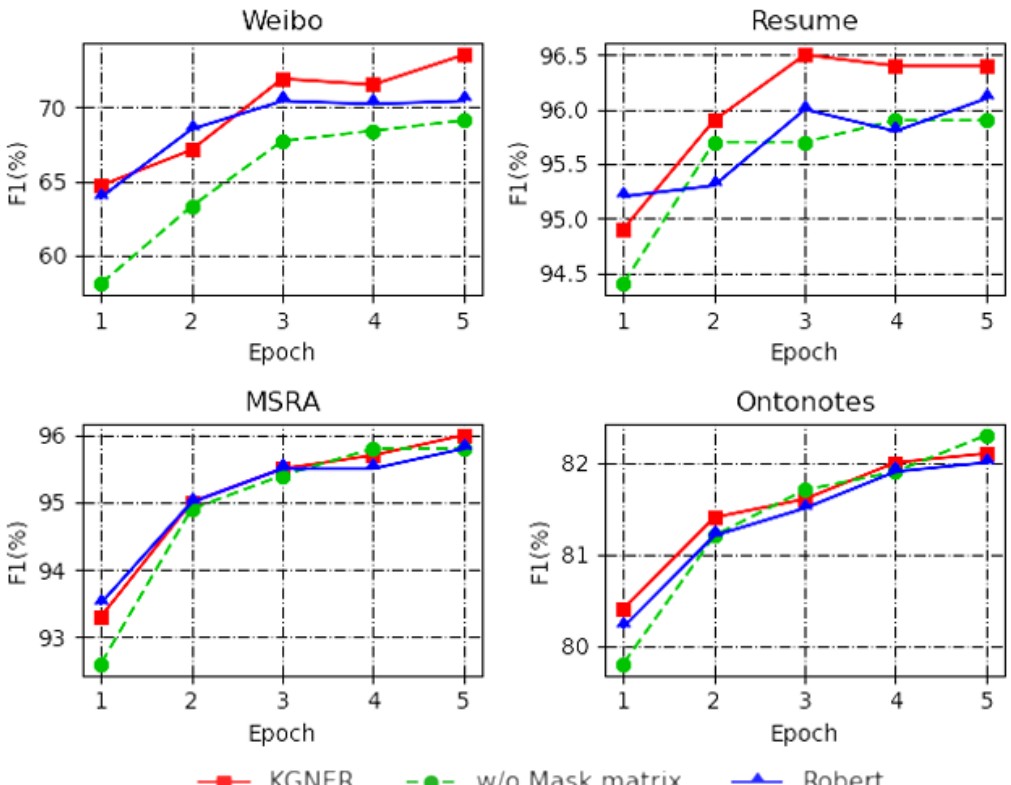

**Figure 5.** Ablation study on the four datasets.

### 5.2. Performance against Adding Different Knowledge

The Knowledge Graph contains a large amount of structured data, and there are different ways to make full use of the existing knowledge. Table 3 shows the obtained results by adding different knowledge to the model. Adding the relational knowledge included in triple only provides a modest boost to the Ontonotes dataset. However, on the Weibo dataset, the $F1$ value drops by more than 5.0. When adding both relationships and tail entities, the $F1$ value gains a little in MSRA datasets. This shows that adding more knowledge to the sentence is not better. We infer that this may be subjected to the different kinds of datasets.

**Table 3.** Adding different knowledge to the sentence ($F1$).

| Knowledge Type | Weibo | Ontonotes | MSRA | Resume |
|:---:|:---:|:---:|:---:|:---:|
| tail entity | 71.9 | 82.1 | 95.7 | 96.4 |
| relation | 66.6 | 82.2 | 95.8 | 96.1 |
| both | 72.0 | 82.1 | 95.8 | 96.3 |

### 5.3. Performance against Using Different Knowledge Graph

- **CN-DBpedia** [34] is a large open-field encyclopedic KG developed by the Knowledge Work Lab of Fudan University, which involves a large number of entities and relationships. CN-DBpedia has been refined by eliminating those triples with entity names of less than 2 in length or containing special characters. There are 5.17 million triples in the improved CN-DBpedia.
- **HowNet** [35] is a large-scale language knowledge base for Chinese vocabulary and concepts, in which each Chinese word is annotated with semantic units called sememes. If we take (word, contain, sememes) as a triple, HowNet is a language KG. Similarly, the official HowNet is refined by eliminating those triples with entity names less than 2 in length or containing special characters. There are 52,576 triples in the improved HowNet.

- **MedicalKG** is a Chinese medical concept KG developed by [12]. There are four types of hypernym(sysptoms, diseases, parts, and treatments) and 13,864 triples in it.
- **Medicine_NER** is the Clinical Named Entity Recognition (CNER) task released in CCKS 2017.

We use the three different knowledge graphs to perform substantial experiments on five datasets. Table 4 shows the experimental results. Compared with HowNet and MedicalKG, Weibo and Resume achieve the excellent $F1$ score on CN-DBpedia, but Ontonotes obtains the highest $F1$ score on HowNet. We speculate the following reasons: the Weibo and Resume dataset are obtained from the Internet and better match the knowledge in CN-DBpedia; Ontonotes is in the news domain and can acquire more knowledge from the HowNet. Finally, Medicine_NER only achieves the best results for MedicalKG. From the above results, we conclude the correct selection of KG is of great benefit to domain-specific tasks.

**Table 4.** Comparison of different knowledge graphs ($F1$).

| Knowledge Graph Type | Weibo | Ontonotes | MSRA | Resume | Medicine_NER |
|---|---|---|---|---|---|
| HowNet | 70.4 | 82.6 | 95.9 | 96.2 | 93.8 |
| CN-DBpedia | 71.9 | 82.1 | 95.7 | 96.4 | 93.8 |
| MedicalKG | 68.7 | 82.1 | 95.9 | 96.2 | 94.1 |

*5.4. F1 Score against Sentence Length*

Based on the sentence length, we divided the test dataset into six parts. The second column denotes the number of sentences corresponding to each length range. The third column counts the number of sentences that can be matched to knowledge. The fourth column is the ratio of the two left columns. Table 5 can reflect some of the following information. First, the performance of both short and long sentences is not very good. Some of the reasons are as follows. The number of short sentences that can be matched to knowledge is too low. Although long sentences can match increased knowledge, the sentence after adding the knowledge is too long, increasing the semantic complexity of the sentence. In contrast, sentences from 40 to 100 in length not only match a lot of knowledge but also lead to a higher $F1$ score.

**Table 5.** $F1$ score against sentence length on the OntoNotes dateset.

| Sentence Length | Sentence Number | Matched Number | Proportion (%) | KGNER/BERT ($F1$) |
|---|---|---|---|---|
| $l < 20$ | 1685 | 153 | 9.0 | 77.5/73.0 |
| $20 \leq l < 40$ | 1302 | 413 | 31.7 | 81.2/75.9 |
| $40 \leq l < 60$ | 798 | 360 | 45.1 | 82.7/75.9 |
| $60 \leq l < 80$ | 365 | 232 | 63.5 | 84.3/77.5 |
| $80 \leq l < 100$ | 169 | 103 | 60.9 | 80.0/73.9 |
| $100 \leq l$ | 124 | 92 | 74.1 | 82.6/77.7 |

*5.5. Efficiency Comparison*

As shown in Figure 6, we compared the inference speed of the three models on four datasets. As we can see, on different datasets, the inference time becomes longer as the dataset size increases, in addition to Ontonotes dataset. Using the same data, the inference time of both KBERT and KGNER increases due to the introduction of external knowledge. This may be the reason that the introduced knowledge leads to an increase in the sentence length. Compared with KBERT, except for the Ontonotes dataset, our model inference is a little slower. We infer that this may suffer from the impact of the CRF.

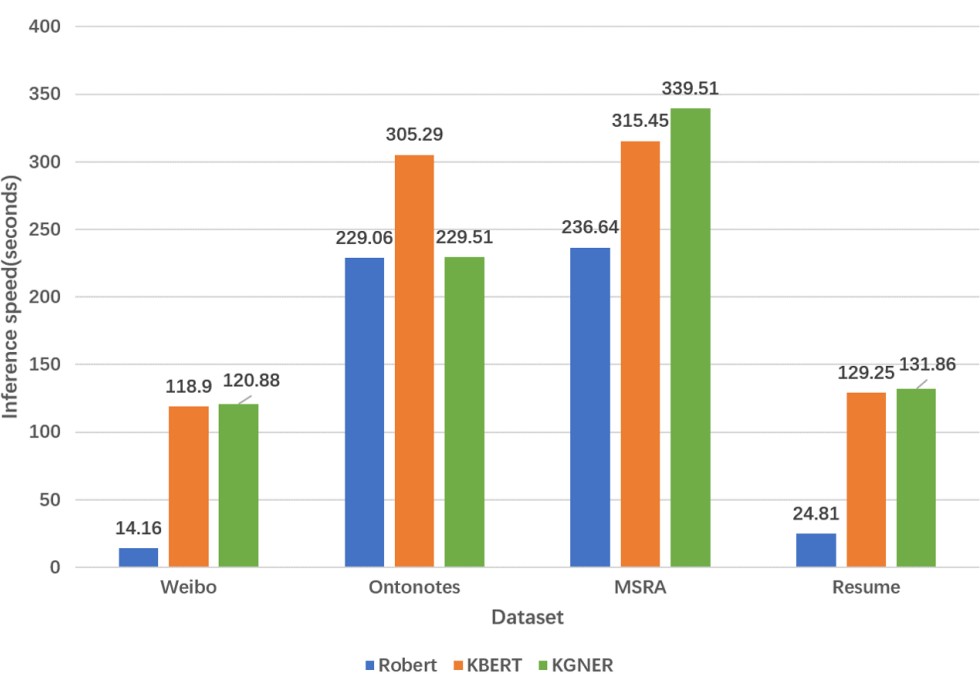

**Figure 6.** Inference speed on the four datasets.

*5.6. Case Study*

Table 6 illustrates examples of Chinese NER tagging results on Ontonotes and Weibo datasets, respectively. For a sentence, "*The old woman won twelve games in the match*". "老妇人 (The old woman)", the nickname of a football club Juventus F.C., according to our normal understanding, if matched knowledge is introduced, "老妇人 (The old woman)" should be classified as "ORG". However, the results were contrary to expectations. It is not even correctly labeled on the given datasets. Nevertheless, this is difficult to understand, since we do not introduce domain-specific knowledge in the training phase. Consequently, it is indispensable to introduce domain-specific knowledge according to different tasks.

**Table 6.** Labeling comparison on different datasets.

| Sentence | The old woman won twelve games in the match.<br>老 妇 人 二 月 赢 了 十 二 场 比 赛 |
|---|---|
| Matched knowledge | {老妇人 (The old woman), 类别 (Type), 足球俱乐部 (Football club)} |
| Gold label | 老 (B-ORG) 妇 (M-ORG) 人 (E-ORG) 二 (O) 月 (O) 赢 (O) 了 (O) 十 (O) 二 (O) 场 (O) 比 (O) 赛 (O) |
| Ontonotes label | 老 (O) 妇 (O) 人 (O) 二 (O) 月 (O) 赢 (O) 了 (O) 十 (O) 二 (O) 场 (O) 比 (O) 赛 (O) |
| Weibo label | 老 (B-PER) 妇 (M-PER) 人 (E-PER) 二 (O) 月 (O) 赢 (O) 了 (O) 十 (O) 二 (O) 场 (O) 比 (O) 赛(O) |

**6. Conclusions**

In this paper, we proposed KGNER, a knowledge graph-inspired, named-entity recognition model, aiming to incorporate knowledge into NER. To prevent the introduced knowledge from diverting the sentence from its correct meaning, we conceived a new means of position encoding for raw sentences and knowledge and subjected the knowledge's impact to the masking matrix. This not only preserves the original information of the sentence but also avoids the differentiation caused by the different use of vector space. Compared to baseline, since we use CRF and build the sentence tree, our model contains a slight time delay. Nevertheless, the experimental results show that our model outperforms other lexicon-based models in four Chinese datasets. It also shows that introducing

knowledge into NER is a promising endeavor. We will continue to explore the potential of KGNER on other NLP tasks as future work.

**Author Contributions:** Conceptualization, W.H. and L.H.; methodology, W.H.; formal analysis, H.M.; investigation, H.M.; resources, K.W.; data curation, J.X.; writing—original draft preparation, W.H.; writing—review and editing, L.H.; All authors have read and agreed to the published version of the manuscript.

**Funding:** This research received no external funding.

**Institutional Review Board Statement:** Not applicable.

**Informed Consent Statement:** Not applicable.

**Data Availability Statement:** The Weibo, Resume, MSRA and Ontonotes datasets are publicly available at: https://github.com/liuwei1206/LEBERT. The HowNet, CN-DBpedia, MedicaKG, and Medicine_NER are publicly available at: https://github.com/autoliuweijie/K-BERT.

**Acknowledgments:** We sincerely appreciate the anonymous reviewers for their precious comments and valuable suggestions. Moreover, we genuinely thank my teacher for his constructive guidance for the period of production of this paper and my friends Xuebin Jing and Zhenping Kang for their help polishing our paper.

**Conflicts of Interest:** The authors declare no conflict of interest.

## Abbreviations

The following abbreviations are used in this manuscript:

| | |
|---|---|
| NER | Named entity recognition |
| BERT | Bidirectional Encoder Representation Transformers |
| CRF | Conditional random field |
| NLP | Natural language procession |
| Q&A | Question and answer |
| ELMO | Embeddings from Language Models |
| KG | Knowledge graphs |
| KN | Knowledge noise |
| HF | Heterogeneous information fusion |
| LSTM | Long Short-Term Memory |
| RNN | Recurrent neural network |
| PTMs | Pre-Trained Models |
| GPT | Generative Pre-Training |
| BART | Bidirectional and Auto-Regressive Transformers |
| KB | Knowledge base |
| GRU | Gated recurrent unit |
| KGE | Knowledge graph embedding |
| KL | Knowledge layer |

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
