# Peer review of "KGNER: Improving Chinese Named Entity Recognition by BERT Infused with the Knowledge Graph"

_applsci, doi:10.3390/app12157702_

Round 1

Reviewer 1 Report

This is an interesting and very well-written paper, on named entity recognition (NER).  The authors propose a knowledge graph-inspired named entity recognition (KGNER) featuring as a masking and encoding method to incorporate common sense into bidirectional encoder representations from transformers (BERT). 

They use conditional random field (CRF) as the backend, and show improved overall performance on four datasets.

I must note that my expertise is not directly in NLP, but in speech processing, and so I am not so up-to-date in terms of judging the novelty of this work.

Given the importance and use of F1 in this paper, F1 deserves much more explanation; also, why use it, rather than WER or CER?

Specific points:

..common sense knowledge in  knowledge graph. ->

..common sense knowledge in the knowledge graph. 

(Same in the paper title; add “the” elsewhere in the paper for this phrase, too)

..We proposed a knowledge ..->

..We propose a knowledge ..

(To be consistent with the tense later in the abstract)

..it has also casted a attention to academic .. ->

..it has also cast attention to the academic ..

..Although success results have .. ->

..Although successful results have ..

..this process of traing can .. ->

…this process of training can ..

..two challenges lies in the road .. ->

..two challenges in the road ..

OR

..two challenges that lie in the road ..

..this paper proposes an named .. ->

..this paper proposes a named ..

..results in section 4,5. ->

..results in sections 4 and 5.

..Named entity recognition task has ..

..The named entity recognition task has ..

..pre-trained encoders outputs vectors ..

..pre-trained encoder output vectors ..

..tasks using Fine-Tuning model.

..tasks using a Fine-Tuning model.

..achieve state-of-the-art 152 performance on CWS. - what is CWS?

..used Bart to solve ..

..used BART to solve ..

..Everyone is also actively ..

..Researchers are also actively ..

..Since TransE [36] and other ..

..TransE [36] and other ..

..domain, compared to other model.

..domain, compared to other models.

..sentence with n characters .. - use italics for all math symbols, including those in text like this.

..the i-th character .. - same comment

..name of entities.

..names of entities.

..and a KG. we can ..

..and a KG, we can ..

Figure 3. Flat the sentence tree.  - what does this mean?

..In order not to affect the original sentence’s meaning.  - not a sentence

..An example to explain it.

..Here is an example to explain it.

..adjective is more likely to be followed by a noun than a verb .. - for English, yes; not for French, for example.

..model label sequence jointly .. ->

..model a label sequence jointly ..

In Table 1, I assume you mean “Sentence”; also, why not simply spell out “char” as “character”?

..measures which defined as follows:

..measures that are defined as follows:

..through an extra multi-layers of ..

..through extra multi-layers of ..

..Weibo and Resueme data ..

..Weibo and Resume data ..

..(The old woman)" shoule ..

..(The old woman)" should ..

..not hard to understand. Since ..

..not hard to understand, since ..

..survey. Knowledge and Information systems ..

..survey. Knowledge and Information Systems ..

(i.e., capitalize first letters in titles; same in ref. 23 and likely elsewhere; same for Chinese in ref. 25-26)

Refs. 10-12, 16, 20, 23 etc etc: In Proceedings of the Proceedings of ..  ->

In the Proceedings of ..

ref. 57 has no source

Reviewer 2 Report

This paper presents a new methodology for named entity recognition (NER) in Chinese that combines language models with Knowledge Graphs (KGs). At the theoretical level, the presentation of the proposed method is detailed and complete. It includes the figures and tables necessary to understand the system architecture and the techniques that have been integrated, as well as the details necessary to understand the formulas and calculations that are performed in each process that make up the proposed model. Related work is also extensively covered, as a convenient starting point. References are also numerous, relevant, and up-to-date.

Various experiments are conducted on 4 named entity datasets to compare 13 models, including the proposed model, using accuracy, recall and F1 metrics for evaluation. These are quality datasets and models widely used today, including one of the most effective performance metrics in machine learning. Given the complexity of the proposed model, it would have been desirable to provide a comparison of response times with the same models and datasets proposed for performance testing. In addition to comparing these methods, variants of the proposed model have also been compared by eliminating some of the elements that compose the model. On the other hand, two general domain KGs in Chinese have also been compared, which offers a point of view that is also of interest for evaluating the proposed model. The presentation of the results is correct, including tables and graphs. It should be noted however that the graphs shown in figure 5 are difficult to understand at first glance because the legend is not placed in a sufficiently visible and clear area. On the other hand, it is indicated in the text itself that the system could perform better with domain-specific KGs, and a specific medical domain KG (MedicalKG) is cited. However, there is no experiment involving MedicalKG or any other similar KG. This would have offered a broader view of the scope of the proposed model. Some further experiments with other domain-general multilingual KGs such as Wikidata (cf. Van Veen, 2019) or BabelNet (cf. Navigli et al., 2021) are also missing. In addition, latency could be compromised due to the integration of several methods. There are no comments about this possible limitation in Results and discussion.

Finally, the text would benefit from some minor revisions of style, formatting, and spelling. Some examples follow:

- The last sentence of line 269 is disconnected from the following sentence in line 270.

- The sentence on lines 298, 299 and 300 is redundant and repeats a concept that has already been abbreviated/mentioned

- For figure 5 it would be desirable to place the legend in a more visible place.

- Table 6 is not formatted in the same way as the others and is a bit of a clash. It also needs to fit within the margins of the text.

- The example sentence in line 403 should start with "The old woman..." instead of "The old man...".

- In line 404, after "Juventus F.C.," the following word should be lower case.

- In line 405 it says "shoule" and should be "should".

- Between lines 409 and 410 there should be extra space.
